# PAFT: A Parallel Training Paradigm for Effective LLM Fine-Tuning

## Abstract

Large language models (LLMs) have shown remarkable abilities in diverse natural language processing (NLP) tasks. The LLMs generally undergo supervised fine-tuning (SFT) followed by preference alignment to be usable in downstream applications. However, this sequential training pipeline leads to alignment tax that degrades the LLM performance. This paper introduces PAFT, a new **PA**rallel training paradigm for effective LLM **F**ine-**T**uning, which independently performs SFT and preference alignment (e.g., DPO and ORPO, etc.) with the same pretrained model on respective datasets. The model produced by SFT and the model from preference alignment are then merged into a final model by parameter fusing for use in downstream applications. This work reveals important findings that preference alignment like DPO naturally results in a sparse model while SFT leads to a natural dense model which needs to be sparsified for effective model merging. This paper introduces an effective interference resolution which reduces the redundancy by sparsifying the delta parameters. The LLM resulted from the new training paradigm achieved Rank #1 on the HuggingFace Open LLM Leaderboard[1]. Comprehensive evaluation shows the effectiveness of the parallel training paradigm.

## 1 Introduction

In recent years, large language models (LLMs) have emerged as the standard approach to addressing natural language processing (NLP) tasks. The typical way of building an LLM for downstream applications generally follows a sequential training pipeline consisting of two phases: 1. Supervised Fine-tuning (SFT), where the pre-trained LLM is fine-tuned with the language modelling loss on demonstrations of the desired behaviour. 2. Alignment with human preference, where the model produced by the SFT phase is further fine-tuned with an alignment algorithm like Reinforcement Learning from Human Feedback (RLHF) or Direct Preference Optimization (DPO), etc. While this sequential pipeline has been used to seemingly great success, how the SFT and the preference alignment work better with each other is underexplored.

Recent studies OpenAI (2023); Askell et al. (2021); Song et al. (2023) have found that the preference alignment phase can cause the LLM to forget the diverse capabilities that it has acquired from earlier phases, despite aligning the LLM with human expectation. This phenomenon, also known as the *alignment tax* in the literature Ouyang et al. (2022), has accumulated substantial attention from both academia and industry. The alignment tax inherently results from catastrophic forgetting present in the staged training. Independent studies, including DeepSeek R1 DeepSeek-AI et al. (2025), challenge the necessity of sequential pipelines by demonstrating that reinforcement learning paradigms can directly optimize alignment without requiring prior SFT. This finding suggests that SFT and alignment may target orthogonal objectives, motivating alternative approaches like parallel training. To reduce catastrophic forgetting and thus alignment tax, this paper introduces a new parallel training paradigm for LLM fine-tuning, named PAFT, which independently performs SFT and preference alignment with the same pre-trained model on respective datasets, instead of sequentially conducting SFT followed by preference alignment. The model from SFT and the model from

---

[1]`https://huggingface.co/spaces/open-llm-leaderboard-old/open_llm_leaderboard` Uncheck the *private or deleted* option to make our private Rank #1 model visible.

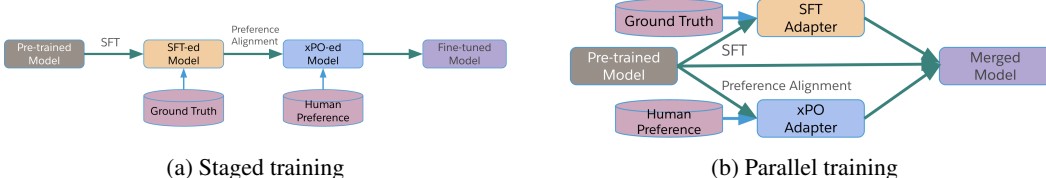

(a) Staged training            (b) Parallel training

Figure 1: Comparison of training paradigms.

preference alignment are then merged into a final model by parameter fusing for use in downstream applications.

As discovered by prior work Yadav et al. (2023); Yu et al. (2023), direct model merging causes the parameter values to interfere across models, thereby harming the performance of the final model. The interference, which reduces parameter magnitudes in the merged model and eliminates subtle distinctions among values, can attribute to the redundant *delta parameters*, i.e., the differences in values between fine-tuned and pre-trained parameters, resulted from fine-tuning. Previous studies on model pruning Hoefler et al. (2021); Thimm & Fiesler (1995) have shown that during fine-tuning, many model parameters can change over the course of fine-tuning but only have a small impact on performance. However, when merging a parameter that is influential for one model but redundant (i.e. not influential) for other models, the influential value may be obscured by the redundant values, lowering the overall model performance. This work reveals the dense properties of the delta parameters resulted from SFT. To mitigate the dense property of SFT, we propose an effective interference resolution which reduces the redundancy by sparsifying the delta parameters by adding a L1-norm penalty to the original SFT loss function. The existing findings indicate that the inclusion of the L1 term enhances the sparsity of the SFT. This method of implicitly inducing sparsity has been evaluated against a technique that introduces sparsity explicitly, i.e., DARE Yu et al. (2023), demonstrating the advantages of employing the L1-norm on LLM's performances in downstream tasks.

Finally, the sparse delta parameters from SFT and preference alignment are merged into a single stronger model. Different merging methods are assessed, and TIES and Task Arithmetic are shown to be the best model merging methods, depending on base models. The method of Parallel SFT$_{sparse}$+DPO merged through TIES based on Mistral-7B sets a new benchmark for 7B models, i.e., 0.6524 on average over the six tasks in HuggingFace Open LLM Leaderboard. Notably, Parallel SFT$_{sparse}$+DPO consistently outperforms Parallel SFT+DPO across all model merging methods, showing the effectiveness and robustness of the PAFT training paradigm.

The contributions of this paper are threefold:

1. Evidence is presented that parallel training of SFT and preference alignment outperforms sequential training, effectively reducing the alignment tax.

2. The significance of sparse model integration is highlighted as a mean to prevent model conflict while preserving the full capability of each model. We demonstrate the superiority of the L1-norm over DARE as a more effective and higher-quality method for promoting sparsity in model training across various model merging techniques.

3. We conduct comprehensive evaluation of PAFT on well-known public benchmarks including Open LLM Leaderboard and AlpacaEval. The PAFT-ed 7B model achieved Rank #1 in the 7B/8B model category on the Open LLM Leaderboard, and the PAFT-ed 70B model topped the Leaderboard globally.

## 2 RELATED WORKS

### 2.1 SFT AND HUMAN PREFERENCE ALIGNMENT

The standard LLM pipeline—pretraining followed by supervised fine-tuning (SFT)—was popularized by models such as BERT Devlin et al. (2019) and GPT-4 OpenAI (2023). SFT boosts downstream accuracy, but unaligned outputs can be unethical, so many systems add a second stage of

preference alignment via RLHF Christiano et al. (2023); Ziegler et al. (2020); Leike et al. (2018). RLHF typically fits a Bradley–Terry reward model on human comparisons Stiennon et al. (2022) and applies PPO Schulman et al. (2017). Lightweight alternatives include DPO, which directly links the reward model to policy gradients Rafailov et al. (2023), and ORPO, which optimizes odds-ratios in a single update Hong et al. (2024). While alignment can impose an "alignment tax" that erodes SFT gains Ouyang et al. (2022), some work reports "alignment bonuses" when RLHF-trained models exceed their SFT baselines Bai et al. (2022). Recent efforts even bypass strict sequencing by jointly optimizing task and preference losses end-to-end DeepSeek-AI et al. (2025); Kreutzer et al. (2018).

## 2.2 Sparsity in LLM Adaptation

Deploying LLMs on edge devices has spurred compression research, notably pruning Han et al. (2015) and LoRA adapters Hu et al. (2022). Surveys confirm that integrating sparsity can cut inference costs without large accuracy drops Zhu et al. (2023). In LoRA, fine-tuned adapter matrices often contain many near-zero weights; pruning a fraction $p$ of these and rescaling the remainder by $1/(1 - p)$ recovers much of the original performance Yu et al. (2023). Alternative approaches impose an $\ell_1$ penalty—akin to Lasso Santosa & Symes (1986) or compressed sensing priors Candes et al. (2006)—to induce sparse adapters during training.

## 2.3 Model Merging

Rather than expensive multi-task training Poth et al. (2021); Wang et al. (2020); Fifty et al. (2021), merging separately fine-tuned models can efficiently combine capabilities. ModelSoup averages weights of SFT checkpoints to achieve SOTA gains Wortsman et al. (2022), and Fisher merging refines this by weighting updates by their Fisher Information Matena & Raffel (2022). Task-arithmetic extends averaging via vector addition/subtraction for analogies or forgetting Ilharco et al. (2023), while RegMean solves per-layer regressions to estimate merged parameters Jin et al. (2023). To reduce destructive interference among tasks, TIES filters by magnitude and enforces consistent update signs Yadav et al. (2023), yielding more robust merged models.

## 3 Methodology

### 3.1 Problem Setting

Given a pre-trained LLM, such as Mistral and Llama, we aim to optimize the model for a wide range of downstream tasks by fine-tuning it either fully or with parameter-efficient tuning such as LoRA Hu et al. (2022), using SFT and preference alignment. Throughout this paper, $\theta$ denotes the trainable parameters; $\theta_{\mathrm{pre}}$ denotes the parameters of the pre-trained model; $\theta_{\mathrm{sft}}$ denotes the parameters of the model fine-tuned with SFT; $\theta_{\mathrm{xpo}}$ denotes the parameters of the model fine-tuned with preference alignment, such as PPO Schulman et al. (2017); Ziegler et al. (2020), DPO Rafailov et al. (2023) and ORPO Hong et al. (2024), etc.; $\delta_{\mathrm{sft}} = \theta_{\mathrm{sft}} - \theta_{\mathrm{pre}}$ denotes the delta parameters between the SFT-ed model and the pre-trained model; and $\delta_{\mathrm{xpo}} = \theta_{\mathrm{xpo}} - \theta_{\mathrm{pre}}$ denotes the delta parameters between the preference-aligned model and the pre-trained model.

### 3.2 Parallel Training

SFT and preference alignment are two distinct methodologies designed to enhance the capabilities of pre-trained LLMs for specific applications. SFT focuses on boosting the performance of LLMs on downstream tasks by fine-tuning them with datasets that closely resemble the target task. This process tailors the model's responses to be more accurate and relevant for a specific use-case. In contrast, preference alignment, such as RLHF, DPO and ORPO, etc., is a methodology that refines a model's outputs based on human preferences. It generally fine-tunes the model on pairs of responses to an input query, one of which is preferred over the other one. Preference alignment uses such feedback signal to guide the model towards generating outputs that align with human expectation and ethical standards. This approach is particularly valuable for addressing the ethical considerations that arise when deploying LLMs in real-world scenarios.

Nowadays, researchers have applied SFT to enhance the performance of LLMs on targeted tasks, and then employed preference alignment to further align the models with human preferences. However, this sequential application of SFT followed by preference alignment has often led to a compromise in task-specific performance - a phenomenon referred to as the alignment tax. This occurs because the distinct objectives of SFT and preference alignment can sometimes be at odds, with the alignment process potentially undoing some of the task-specific optimizations achieved through SFT.

We address the challenge of the alignment tax by a novel approach that involves SFT and preference alignment concurrently using adapter training, such as LoRA Hu et al. (2022). This method takes full advantages and strengths of both SFT and preference alignment without sacrificing performance in either one, i.e., ensuring that the resulting model maintains high performance in downstream tasks while also being aligned with human preferences, thus overcoming the limitations associated with the alignment tax. During the training process specifically, based on the same pre-trained model $\theta_{\text{pre}}$, the two separate adapter parameters, denoted as $\delta_{\text{sft}}$ and $\delta_{\text{xpo}}$, are learned in parallel from downstream ground truth and human preferences, respectively. The proposed PAFT seeks to merge the $\delta_{\text{sft}}$ and $\delta_{\text{xpo}}$ in an effective way of avoiding feature interference. Figure 1 compares the typical staged training pipeline and our parallel training pipeline PAFT.

## 3.3 Sparse Merging

The integration of dense neural network models often results in a suboptimal combined model due to the phenomenon of parameter interference. This challenge has led researchers to explore alternative strategies. Our investigations reveal that by increasing sparsity of a fine-tuned adapter, the performance of merging the adapter with the base model can be improved. Specifically, the parameter $\delta_{\text{xpo}}$, derived from adapter training like LoRA for preference alignment, demonstrates clear sparsity, as depicted in Figure 2. We hypothesize that this sparsity results from the mode-seeking behavior inherent in the constraint optimization objective of preference learning like DPO. For example, DPO includes a KL divergence term, which has been associated with mode-seeking properties based on the type of initialization in prior work on preference optimization Tajwar et al. (2024). Mode-seeking objectives tend to concentrate probability mass on specific, high-reward outputs, potentially leading to more focused and sparse parameter updates.

In contrast, the sparsity in a SFT adapter, denoted by $\delta_{\text{sft}}$, is not pronounced. This can be because SFT's maximum likelihood objective, similar to behavior cloning, attempts to increase the likelihood of all positive examples, potentially resulting in more distributed and dense parameter updates across the adapter. It aligns with the findings of Piao et al. (2022), which showed that maximum likelihood training tends to produce dense representations. To increase the sparsity within $\delta_{\text{sft}}$, we propose the incorporation of an L1 regularization term during the SFT process. This modification to the fine-tuning procedure is expressed mathematically as follows:

$$L_{\text{SFT}_{\text{sparse}}} = L_{\text{SFT}} + \lambda \cdot \|\delta_{\text{sft}}\|_1 \tag{1}$$

Here, $L_{\text{SFT}}$ represents the conventional cross-entropy loss function, and $\lambda$ is a weighting factor that controls the strength of the sparsity regularization. Our results indicate that this approach significantly enhances the sparsity of $\delta_{\text{sft}}$, with sparsity levels over 90%, as illustrated by the SFT_sparse in Figure 2.

Given sparse representations for adapters of both SFT and preference alignment, the challenge is to effectively merge these delta parameters, $\delta_{\text{sft}}$ and $\delta_{\text{xpo}}$, with the original pre-trained model, $\theta_{\text{pre}}$, while preserving the performance benefits of SFT and preference alignment. The merging process can be formalized by the equation:

$$\theta_{\text{merge}} = f(\theta_{\text{pre}}, \delta_{\text{dpo}}, \delta_{\text{sft}}) \tag{2}$$

In our study, we explore a variety of merging methods proposed in the literature, including SLERP, Task Arithmetic, TIES, DARE TIES, and Linear. Detailed discussions of these merging methods are provided in the Related Work section.

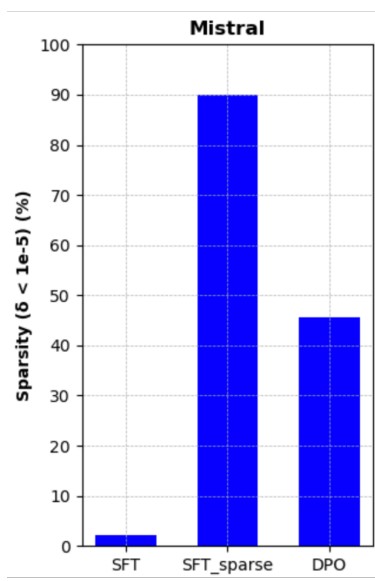 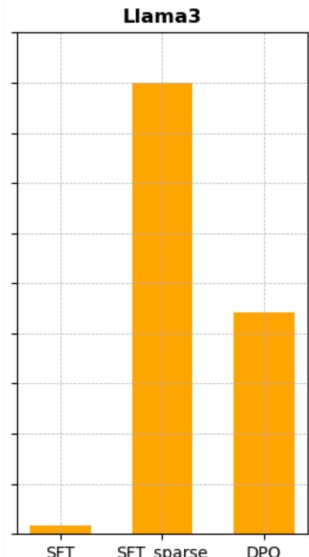

Figure 2: Adapter sparsity for SFT and DPO. The sparsity levels are computed by first merging the parameters from LoRA matrices $\delta_A$ and $\delta_B$ through matrix multiplication ($\delta = \delta_B \times \delta_A$), and computing the percentage of elements within $\delta$ that are less than a threshold of $1 \times e^{-5}$, indicating the proportion of weights approaching zero. The reported sparsity is the average across all layers.

## 4 EXPERIMENTS

### 4.1 EVALUATION SETTINGS

In this study, we conduct comprehensive evaluation on both the Open LLM leaderboard provided by HuggingFace and the AlpacaEval benchmark. The Open LLM Leaderboard benchmark suite encompasses a diverse set of six benchmark tasks, namely ARC, HellaSwag, MMLU, TruthfulQA, Winogrande, and GSM8K, along with their aggregated performance metrics.

In our experiments, we employ two state-of-the-art pre-trained models: Mistral-7B Jiang et al. (2023) and Llama-3-8B Team (2024). This section presents the experimental results of merging the delta parameters obtained through SFT and DPO using the LoRA technique. We also study another preference alignment method ORPO for PAFT, which results in the same observations and conclusions as those from DPO. It shows the generalizability of PAFT to different preference alignment techniques. Due to space limit, we put the experimental results for ORPO in the appendix.

Following the Zephyr work Tunstall et al. (2023), we use the UltraChat Ding et al. (2023) dataset for SFT and the UltraFeedback Tunstall et al. (2023) dataset for DPO. UltraChat is a self-refinement dataset consisting of 200K multi-turn dialogues generated by GPT-3.5-Turbo over 30 topics and 20 different types of text material. UltraFeedback consists of 64k prompts, each of which have four LLM responses that are rated by GPT-4 according to criteria like instruction-following, honesty, and helpfulness.

We meticulously explore a spectrum of merging methods, including SLERP, Task Arithmetic, TIES, DARE-enhanced TIES, and Linear combination. Each of these merging strategies is scrutinized to determine its efficacy in integrating the sparsity-induced parameters from LoRA with the original pre-trained models. The goal is to ascertain which method most effectively preserves the performance enhancements attributed to SFT and DPO, thereby contributing to the advancement of model merging methods in LLM research. For training individual adapters, we have used the same settings as in the *zephyr-7b-beta* development[2]. Our evaluation is conducted using the EleutherAI's LM Evaluation Harness framework Gao et al. (2023). We adhere to the same branch (b281b09) used by

---

[2]https://github.com/huggingface/alignment-handbook/tree/main/recipes/zephyr-7b-beta

| Base Model: Mistral-7B | | | | | | |
|---|---|---|---|---|---|---|
| **Method** | **ARC** | **HellaSwag** | **MMLU** | **TruthfulQA** | **Winograde** | **GSM8K** | ***AVERAGE*** |
| **PAFT (SFT$_{sparse}$+DPO)** | | | | | | |
| SLERP | 0.6391 | 0.8464 | 0.63961 | 0.5123 | 0.794 | 0.4223 | 0.64228 |
| Task Arithmetic | 0.6519 | 0.8477 | 0.63325 | 0.563 | 0.794 | 0.4071 | 0.64949 |
| TIES | 0.6519 | 0.8551 | 0.63927 | 0.5453 | 0.7946 | 0.4284 | **0.65243** |
| DARE TIES | 0.6493 | 0.8526 | 0.63444 | 0.5454 | 0.7964 | 0.4094 | 0.64792 |
| Linear | 0.6348 | 0.8451 | 0.64275 | 0.505 | 0.7932 | 0.4246 | 0.64091 |
| **Parallel SFT+DPO** | | | | | | |
| SLERP | 0.6391 | 0.8479 | 0.63937 | 0.5031 | 0.7924 | 0.4124 | 0.63904 |
| Task Arithmetic | 0.651 | 0.851 | 0.62998 | 0.5397 | 0.8011 | 0.4117 | 0.64741 |
| TIES | 0.5956 | 0.8319 | 0.61651 | 0.3993 | 0.7853 | 0.3071 | 0.58928 |
| DARE TIES | 0.5922 | 0.8244 | 0.60471 | 0.3801 | 0.7577 | 0.2767 | 0.57263 |
| Linear | 0.6391 | 0.846 | 0.63935 | 0.4946 | 0.7995 | 0.4314 | 0.64166 |
| **Sequential** | | | | | | |
| SFT$_{sparse}$+DPO | 0.6391 | 0.8464 | 0.63461 | 0.4403 | 0.7894 | 0.4123 | 0.62702 |
| SFT+DPO | 0.656 | 0.8459 | 0.62634 | 0.4479 | 0.7884 | 0.3836 | 0.62469 |
| Mistral-7B | 0.6049 | 0.8320 | 0.6369 | 0.4259 | 0.7814 | 0.37 | 0.6085 |

| Base Model: Llama-3-8B | | | | | | |
|---|---|---|---|---|---|---|
| **Method** | **ARC** | **HellaSwag** | **MMLU** | **TruthfulQA** | **Winograde** | **GSM8K** | ***AVERAGE*** |
| **PAFT (SFT$_{sparse}$+DPO)** | | | | | | |
| SLERP | 0.6067 | 0.8367 | 0.66995 | 0.5297 | 0.7837 | 0.5095 | 0.65604 |
| Task Arithmetic | 0.6118 | 0.8411 | 0.66858 | 0.5552 | 0.7806 | 0.5208 | **0.66301** |
| TIES | 0.6101 | 0.8414 | 0.67098 | 0.5313 | 0.7891 | 0.5185 | 0.66023 |
| DARE TIES | 0.6067 | 0.8398 | 0.66945 | 0.5232 | 0.7885 | 0.5163 | 0.65732 |
| Linear | 0.6049 | 0.8329 | 0.67059 | 0.5168 | 0.7837 | 0.5011 | 0.65166 |
| **Parallel SFT+DPO** | | | | | | |
| SLERP | 0.6152 | 0.8347 | 0.66248 | 0.5149 | 0.7869 | 0.5171 | 0.65521 |
| Task Arithmetic | 0.6254 | 0.837 | 0.66089 | 0.5266 | 0.7869 | 0.5133 | 0.65835 |
| TIES | 0.5879 | 0.8092 | 0.65863 | 0.4283 | 0.7545 | 0.4291 | 0.61127 |
| DARE TIES | 0.6007 | 0.8061 | 0.65702 | 0.4233 | 0.7609 | 0.4049 | 0.60882 |
| Linear | 0.6152 | 0.8331 | 0.66614 | 0.5082 | 0.7845 | 0.5095 | 0.65277 |
| **Sequential** | | | | | | |
| SFT$_{sparse}$+DPO | 0.5648 | 0.7984 | 0.62204 | 0.4049 | 0.7766 | 0.3692 | 0.58932 |
| SFT+DPO | 0.5623 | 0.7976 | 0.62258 | 0.4057 | 0.7719 | 0.3662 | 0.58771 |
| Llama-3-8B | 0.5547 | 0.7909 | 0.61603 | 0.3991 | 0.7619 | 0.3687 | 0.58189 |

Table 1: Results of compared methods on the six Open LLM benchmark tasks

the HuggingFace Open LLM Leaderboard Beeching et al. (2023), and evals are run with batch size 1 on an A100 GPU. The hyper parameter $\lambda$ in Equation 1 controls the sparsity of $\delta_{sft}$. Empirical values 0.0001 and 0.001 are validated in our experiments to achieve reasonable sparsity.

## 4.2 PARALLEL TRAINING VS. SEQUENTIAL TRAINING

To demonstrate the advantages of parallel training PAFT, we conducted empirical comparison of parallel and sequential training approaches on the six benchmark tasks using the two pre-trained models: Mistral-7B and Llama-3-8B. The results are given in Table 1. In the Mistral-7B model section, we firstly evaluated the sequential training of SFT followed by DPO, which gave average scores of 0.62. The scores surpass that of the Mistral-7B base model, setting the stage for a comparison with parallel training outcomes.

Furthermore, we performed side-by-side evaluations of SFT$_{sparse}$+DPO training in both parallel and sequential manners. The findings indicate that training SFT with L1 regularization alongside DPO in parallel leads to a performance metric of 0.65 when merging with the TIES method, over 4% higher than the score achieved by training SFT$_{sparse}$ and DPO in sequence. This outcome can be explained by a notable drawback of sequential training which is its tendency to overlook much of the knowledge gained during the SFT stage, suggesting a suboptimal use of SFT data. In contrast, parallel training effectively combines the benefits from SFT and DPO by processing them concurrently. The benefits are mostly preserved during model merging, ensuring efficient utilization of both SFT

| LLM | ARC | HellaSwag | MMLU | TruthfulQA | Winograde | GSM8K | *AVERAGE* |
|---|---|---|---|---|---|---|---|
| **PAFT (Ein-70B)** | 0.7986 | 0.9149 | 0.7805 | 0.7514 | 0.8777 | 0.7544 | **0.8129** |
| Mixtral-8x22B-Instruct | 0.727 | 0.8908 | 0.7777 | 0.6814 | 0.8516 | 0.8203 | 0.7915 |
| Llama-3-70B-Instruct | 0.7142 | 0.8569 | 0.8006 | 0.6181 | 0.8287 | 0.8544 | 0.7788 |
| **PAFT (TextBase-7B)** | 0.7389 | 0.9027 | 0.6478 | 0.7813 | 0.8603 | 0.6793 | **0.7684** |
| Cohere-Command-R+ | 0.7099 | 0.8856 | 0.7573 | 0.563 | 0.854 | 0.7074 | 0.7462 |
| DBRX-132B-Instruct | 0.6783 | 0.8885 | 0.7372 | 0.6702 | 0.8208 | 0.6732 | 0.7447 |
| OpenChat-3.5 | 0.6604 | 0.8293 | 0.6504 | 0.519 | 0.8177 | 0.6816 | 0.693 |
| Llama-3-8B-Instruct | 0.6075 | 0.7855 | 0.6707 | 0.5165 | 0.7451 | 0.6869 | 0.6687 |
| Mistral-7B-Instruct-v0.2 | 0.6314 | 0.8488 | 0.6078 | 0.6826 | 0.7719 | 0.4003 | 0.6571 |
| Gemma-7B | 0.6109 | 0.8247 | 0.6603 | 0.4491 | 0.7845 | 0.5277 | 0.6429 |

Table 2: Comparison with state-of-the-art LLMs on Open LLM Leaderboard (All the scores are obtained from the Leaderboard.)

and DPO data. Our work underscores the enhanced efficacy of the parallel training approach PAFT, which not only maintains the distinct advantages of SFT and DPO, but also outperforms these techniques when they are used sequentially. This finding is confirmed by the evaluation results of using Llama-3-8B as a base model.

### 4.3 SPARSE MERGING VS. DENSE MERGING

Our study has demonstrated the advantages of incorporating sparsity into fine-tuned models. In the context of sequential training, the inclusion of L1 regularization has yielded a modest yet notable improvement. Specifically, in the Mistral-7B section of Table 1, the average score for the sequential $SFT_{sparse}$+DPO stands at 0.627, surpassing the sequential SFT+DPO without L1 regularization, with a score of 0.625. Although the improvement is marginal, it underscores the value of integrating the L1-norm to induce sparsity.

The impact of sparsity becomes more pronounced when examining parallel training scenarios. Across all considered model merging techniques, Parallel $SFT_{sparse}$+DPO, i.e., PAFT, consistently outperforms its counterpart without L1 regularization, Parallel SFT+DPO, thereby highlighting the efficacy of the sparsity induced by L1-norm. Notably, in the case of the TIES and DARE TIES merging methods, the average score disparity is significant. With TIES, PAFT ($SFT_{sparse}$+DPO) achieves a score of 0.6524, while Parallel SFT+DPO without sparsification lags behind at 0.5893. Similarly, for DARE TIES, PAFT ($SFT_{sparse}$+DPO) scores 0.6479, outstripping Parallel SFT+DPO's 0.5726. This substantial margin illustrates the robustness of L1-norm sparsity for various merging methods.

The same insights as given in the Mistral-7B section can be gained from the Llama-3-8B section in Table 1. PAFT on Llama-3-8B significantly outperforms Parallel SFT+DPO and sequential training. The experimental results confirm the generalizability of PAFT to various pre-trained models.

When comparing different model merging strategies, TIES generally performs better than other methods do on both Mistral-7B and Llama-3-8B, exhibiting superior performance over DARE TIES. DARE, which stands for "Drop And REscale", is a method that explicitly increases sparsity by eliminating elements below a certain threshold and rescaling the remaining parameters. In contrast, the L1-norm introduces sparsity implicitly by integrating it into the objective function. Consequently, the impact of the eliminated terms is less pronounced in the final results compared to DARE. This comparison reveals the advantages of the L1-norm's explicit sparsity induction over the implicit approach employed by DARE.

### 4.4 COMPARISON WITH STATE-OF-THE-ART LLMS

On the online Open LLM Leaderboard, we performed PAFT on the Neurotic-7B[3] and MoMo-70B[4] base models. The two PAFT-ed models significantly improved over the respective base models, and achieved Rank #1 in the 7B/8B model category and globally on the online Open LLM Leaderboard[5],

---

[3] https://huggingface.co/liminerity/Neurotic-Jomainotrik-7b-slerp

[4] https://huggingface.co/leejunhyeok/MoMo-70B-LoRA-V1.2_1

[5] https://huggingface.co/spaces/open-llm-leaderboard-old/open_llm_leaderboard Uncheck the *Private or deleted* option to make our private Rank #1 model visible.

| LLM | LC WinRate | WinRate |
|---|---|---|
| GPT-4 Preview | 50.0% | 50.0% |
| Claude 3 Opus | 40.5% | 29.1% |
| **PAFT 70B** | 38.6% | 26.5% |
| GPT-4 (03/14) | 35.3% | 22.1% |
| Claude 3 Sonnet | 34.9% | 25.6% |
| Llama 3 70B Instruct | 34.4% | 33.2% |
| Mixtral 8x22B v0.1 | 30.9% | 22.2% |
| **PAFT 7B** | 30.6% | 22.8% |
| DBRX Instruct | 25.4% | 18.4% |
| Mixtral 8x7B v0.1 | 23.7% | 18.3% |
| Llama 3 8B Instruct | 22.9% | 22.6% |
| GPT 3.5 Turbo | 22.7% | 14.1% |
| Mistral 7B v0.2 | 17.1% | 14.7% |

Table 3: Comparison with state-of-the-art LLMs on AlpacaEval benchmark using GPT-4 as a judge

| $\lambda$ | **1.0** | **0.1** | **0.01** | **0.001** | **0.0001** | **0.00001** |
|---|---|---|---|---|---|---|
| | 0.6354 | 0.6408 | 0.6488 | **0.6524** | 0.6522 | 0.6505 |

Table 4: Performance ablation with various L1 regularization strengths ($\lambda$). Higher $\lambda$ increases adapter sparsity but can degrade task performance if taken to an extreme.

respectively, showing the effectiveness of PAFT on various base models. Table 2 gives the results of our PAFT-ed models and the existing state-of-the-art models on the Leaderboard.

Additionally, we compared the two PAFT-ed models with existing state-of-the-art LLMs on the AlpacaEval benchmark Li et al. (2023), where every model generates responses to 805 questions on different topics, mostly focused on helpfulness. The models are judged by GPT-4, and the final metric is the pairwise win-rate against GPT-4. As shown in Table 3, the PAFT-ed 70B model outperforms existing state-of-the-art LLMs, except *GPT-4 Preview* and *Claude 3 Opus* in LC (Length-controlled) Win-Rate. While the GPT-4 judge favors its own GPT model family, the PAFT-ed 70B model performs better than *GPT-4 (03/14)* and *GPT 3.5 Turbo* do. On the other hand, the PAFT-ed 7B model outperforms all the 7B/8B and smaller models on AlpacaEval. It even beats some larger models, such as *DBRX Instruct* and *Mixtral 8x7B*.

## 5 ABLATION STUDIES

### 5.1 ROLE OF SPARSIFICATION FACTOR $\lambda$

We previously showed that sparsifying the SFT adapter with an L1 penalty (i.e., $\lambda\|\delta_{\text{sft}}\|_1$) helps reduce parameter interference when merging with DPO. Here, we extend our ablation by varying $\lambda$ over a wider range. Table 4 shows the average downstream performance on the same six Open LLM benchmark tasks for *Parallel* SFT$_{\text{sparse}}$ + DPO using TIES merging under different $\lambda$ values. We observe that $\lambda = 0.001$ provides the best trade-off between sparsity and downstream performance.

### 5.2 EVALUATION ON ALPACAEVAL

Beyond the six classification-style tasks used by the Open LLM Leaderboard, we further evaluated different training strategies on the AlpacaEval benchmark Li et al. (2023). Table 5 shows that our parallel training (PAFT) consistently surpasses sequential training in both the 7B and 70B model size categories, indicating improved alignment and response quality. The parallel 70B model not only outperforms the sequential 70B baseline, but also surpasses older snapshots of GPT-4 in some cases. These results support our main claim that *parallel* training successfully retains both the specialized skills from SFT and the alignment benefits from DPO while mitigating catastrophic forgetting.

| LLM | LC WinRate | WinRate |
|---|---|---|
| GPT-4 Preview | 50.0% | 50.0% |
| Claude 3 Opus | 40.5% | 29.1% |
| **PAFT 70B** | 38.6% | 26.5% |
| Sequential SFT+DPO 70B | 36.2% | 24.0% |
| DPO-alone 70B | 35.5% | 23.1% |
| GPT-4 (03/14) | 35.3% | 22.1% |
| **PAFT 7B** | 30.6% | 22.8% |
| Sequential SFT+DPO 7B | 26.5% | 19.3% |
| DPO-alone 7B | 24.3% | 18.1% |
| SFT-alone 7B | 18.8% | 17.0% |

Table 5: Evaluation on AlpacaEval benchmark. "LC WinRate" denotes the length-controlled win rate against GPT-4, while "WinRate" is the normal pairwise win rate.

## 6 CONCLUSION

LLM fine-tuning generally undergoes a two-stage training process, with SFT applied initially, followed by preference alignment. Yet, research indicates that this sequential approach incurs an "alignment tax", compromising the LLM's overall performance. To counteract this, we advocate for a parallel training strategy PAFT which preserves the advantages of both SFT and preference alignment without incurring the alignment tax associated with sequential training. A significant hurdle in parallel training is the potential for conflict during the model merging phase, where the merging of different adapters can lead to diminished performance. In this paper, we propose the integration of an L1 regularization to the training loss during the SFT phase to induce sparsity, thereby reducing interference between models.

Our experimental results demonstrate the efficacy of incorporating an L1-norm into the SFT process for sparsification and utilizing a parallel training framework over the typical sequential approach. When combining all of them together, i.e. Parallel SFT$_{sparse}$+DPO achieves the state-of-art results on both the LLM leaderboard by HuggingFace and the AlpacaEval benchmark. The ORPO experimental results given in the appendix show the same patterns, demonstrating the generalizability of our PAFT to various preference alignment methods. This comprehensive strategy highlights how the methods of integrating SFT with preference alignment can greatly enhance LLM fine-tuning. Despite its effectiveness, the parallel training process is somewhat cumbersome, requiring two distinct stages: training SFT and DPO in parallel and then merging them together. A more streamlined approach that integrates SFT and DPO training while preserving the benefits of both methods in a single stage is highly desirable.

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

| | Base Model: Meta-Llama-3-8B | | | | | | |
|---|---|---|---|---|---|---|---|
| **Method** | **ARC** | **HellaSwag** | **MMLU** | **TruthfulQA** | **Winograde** | **GSM8K** | *AVERAGE* |
| **PAFT (SFT$_{sparse}$+ORPO)** | | | | | | | |
| SLERP | 0.599 | 0.8217 | 0.665 | 0.4926 | 0.7845 | 0.4898 | 0.6421 |
| Task Arithmetic | 0.5964 | 0.8214 | 0.6655 | 0.4995 | 0.783 | 0.4814 | 0.6412 |
| TIES | 0.5947 | 0.8226 | 0.66358 | 0.4931 | 0.783 | 0.4852 | 0.64036 |
| DARE TIES | 0.593 | 0.8224 | 0.6637 | 0.4921 | 0.783 | 0.4738 | 0.638 |
| Linear | 0.5964 | 0.8206 | 0.6654 | 0.4923 | 0.7814 | 0.4905 | 0.6411 |
| **Parallel SFT+ORPO** | | | | | | | |
| SLERP | 0.6049 | 0.8227 | 0.668 | 0.4905 | 0.783 | 0.4951 | 0.644 |
| Task Arithmetic | 0.6152 | 0.8209 | 0.6621 | 0.4908 | 0.7845 | 0.4989 | 0.6454 |
| TIES | 0.593 | 0.8139 | 0.6633 | 0.4446 | 0.768 | 0.467 | 0.6250 |
| DARE TIES | 0.5981 | 0.8101 | 0.66 | 0.4398 | 0.7632 | 0.4534 | 0.6208 |
| Linear | 0.6067 | 0.8222 | 0.6685 | 0.4868 | 0.783 | 0.4989 | 0.6444 |
| **Sequential** | | | | | | | |
| SFT$_{sparse}$+ORPO | 0.5563 | 0.8018 | 0.62116 | 0.4068 | 0.7719 | 0.3662 | 0.58736 |
| SFT+ORPO | 0.5589 | 0.8021 | 0.62142 | 0.4092 | 0.7711 | 0.3677 | 0.5884 |
| Llama-3-8B | 0.5547 | 0.8009 | 0.61854 | 0.3991 | 0.7619 | 0.3587 | 0.58231 |

Table 6: Results of compared methods with ORPO on the six benchmark tasks

# A  APPENDIX

## A.1  PAFT GENERALIZATION TO DIFFERENT PREFERENCE OPTIMIZATION ALGORITHMS

To demonstrate the generality of *parallel* training with other preference-alignment algorithms, we validated our approach using ORPO Hong et al. (2024), which directly optimizes preferences between two candidate responses in single stage. Table 6 reports the results of combining SFT and ORPO in parallel (denoted as *Parallel SFT+ORPO* and *Parallel SFT$_{sparse}$+ORPO*) on a Llama-3-8B base model. We observe the same pattern of parallel training consistently outperforming sequential training.

Table 6 shows experimental results with ORPO as the preference alignment method alongside SFT with the Llama-3-8B base model. We observe a similar trend where finetuning the LLM sequentially via SFT followed by ORPO underperforms all the parallelly trained variants. Even simple model merging methods such as Task Arithmetic and Linear merging perform strongly, outperforming more complicated methods like DARE TIES in both experiment settings.

These findings confirm that the proposed *Parallel* SFT + preference alignment strategy is not specific to DPO and generalizes effectively to other preference-based algorithms.

## A.2  DISCUSSION

There are a couple of limitations of the parallel training of SFT and preference alignment. Firstly, we have found that sparsity aids in model merging, though the reasons behind this benefit and why DPO initially induces sparsity in the adapter remain unanswered.

Moreover, sparsity can reduce model interference during merging, but the scalability of this approach is still in question. If a merged model deployed in production fails in some cases, it is underexplored how to improve the model responses in these cases. Directly performing SFT on the merged model may lead to catastrophic forgetting of what it learned earlier. On the other hand, parallel training necessitates merging a new SFT-ed model with the existing merged model, adding complexity to the process.

The primary risk associated with this paper pertains to its data usage. Currently, UltraChat data is employed for SFT, while UltraFeedback data is used for preference alignment. UltraChat consists solely of multi-round dialogue data, which inherently limits its format diversity. To enhance the robustness and applicability of the model, it is crucial to incorporate a wider variety of data types beyond dialogue data. Additionally, UltraFeedback relies on annotations generated by GPT-4, which inevitably include errors and inaccurate feedback. To mitigate these risks, higher-quality datasets are needed in the future.

