# OpenReview forum: "PAFT: A Parallel Training Paradigm for Effective LLM Fine-Tuning"
_ICLR.cc/2026/Conference — Submitted to ICLR 2026_

### Official Review · Reviewer_mtHb · 2025-10-27

**Soundness:** 2
**Presentation:** 1
**Contribution:** 2
**Rating:** 2
**Confidence:** 4

**Summary:**

This work introduces PAFT, a parallel training framework for large language models that concurrently fine-tunes supervised (SFT) and preference alignment (e.g., DPO) components, rather than in sequence. The resulting models are then merged via sparse adapters to mitigate the performance degradation known as "alignment tax."

**Strengths:**

1. It shows DPO naturally sparsifies models and introduces L1 regularization to sparsify SFT adapters for cleaner merging.

**Weaknesses:**

1. The paper's organization and clarity could be significantly improved. The methodology section would benefit from a more structured presentation, particularly in explaining the merging process and sparsity mechanisms. Additionally, the flow between technical concepts and experimental results could be smoother to enhance reader comprehension.


2. The reported baseline performance for Llama-3-8B on GSM8K (36.87%) appears anomalously low compared to established results for this model. This discrepancy warrants further investigation or clarification, as it may affect the interpretation of relative improvements claimed by PAFT.

3. The discussion of sparsity in alignment training should be expanded to include relevant prior work. For instance, [1] and other studies have previously explored how reinforcement learning objectives can induce sparsity in LLMs. Engaging with this existing literature would better contextualize the paper's contributions regarding natural sparsity in preference alignment.

4. The work mentions 'alignment tax' but there is no specific analysis how the proposed method solves this issure.


[1] Reinforcement Learning Finetunes Small Subnetworks in Large Language Models, Mukherjee et al.

**Questions:**

See the above section.

---

### Official Review · Reviewer_qVn5 · 2025-10-29

**Soundness:** 3
**Presentation:** 2
**Contribution:** 2
**Rating:** 4
**Confidence:** 3

**Summary:**

This paper introduces PAFT, a novel parallel training paradigm for fine-tuning Large Language Models (LLMs) that aims to mitigate the "alignment tax", which is a phenomenon where the performance of an LLM on downstream tasks degrades after preference alignment. Instead of the conventional sequential approach of Supervised Fine-Tuning (SFT) followed by preference alignment (e.g., DPO), PAFT proposes to perform these two steps in parallel on the same pre-trained model. The resulting models, one from SFT and one from preference alignment, are then merged. The results demonstrate that PAFT outperforms the sequential training paradigm and that the proposed sparse SFT model leads to better merging performance.

**Strengths:**

1. Novel and interesting paradigm: The core ideas of parallelizing SFT and preference alignment are very different from the current mainstream approach of sequential fine-tuning. This is a conceptually interesting and well-motivated approach.
2. Insightful Analysis of Sparsity: The paper provides a valuable insight into the nature of parameter updates during fine-tuning, highlighting the inherent sparsity of DPO and the density of SFT. The proposed technique of enhancing the sparsity of SFT via L1 regularization for improved model merging is a simple yet effective technique. Ablation studies effectively demonstrate the advantages of this approach.

**Weaknesses:**

1. Increased complexity and practicality: While effective, the PAFT paradigm is also more "cumbersome," as discussed in the conclusion. It requires training two separate models and then performing a merge step, which increases the complexity of the process and can be computationally more expensive than sequential approaches. A discussion of the trade-offs in terms of computational resources and engineering overhead would be beneficial.
2. Lack of Direct Quantification of Alignment Tax: While the alignment tax is a primary motivation, the paper misses an opportunity to empirically quantify it within its own experimental setup. A confirmatory study, such as an ablation that explicitly measures the performance drop on downstream tasks after applying preference alignment, would have provided a concrete baseline. This would have more directly demonstrated the magnitude of the problem that PAFT is designed to solve, thereby strengthening the paper's core premise.
3. Limited Theoretical Justification: The success of PAFT depends on the effectiveness of model merging techniques. While this paper evaluates multiple methods, their performance may vary. The paper lacks theoretical discussion on how to choose a merging method. Similarly, the paper notes that the reason for DPO's inherent sparsity is "unanswered", which points to a gap in the current understanding.
4. Limited Experimental Scope: While the empirical evaluation is strong on the chosen benchmarks, its scope could be broader. Including more diverse datasets, RLHF methods, and model architectures would strengthen the generalizability of the findings.

**Questions:**

1. Could you elaborate on the computational costs of PAFT (e.g., total FLOPs, training time, memory) compared to the standard sequential fine-tuning pipeline? Is the overall cost significantly higher, or are there efficiencies to be gained?
2. Table 5 contains an indirect comparison of the quantification of alignment tax. However, DPO-alone performs better than SFT-alone, and no performance drop is observed on downstream tasks after applying preference alignment. Does this experimental result conflict with the core hypothesis?
3. The choice of the L1 regularization hyperparameter λ seems crucial for the success of sparse SFT. The ablation in Table 4 is helpful, but could you provide more details on how the search for the optimal λ was conducted? Was it a simple grid search, and how sensitive is the final performance to small variations around the optimal value?
4. Have you considered other methods for inducing sparsity in the SFT adapter beyond L1 regularization, such as structured pruning or other regularization techniques? It would be interesting to know if the benefits are specific to the L1-norm or if other sparsity-inducing methods would yield similar results.

---

### Official Review · Reviewer_LzpG · 2025-11-01

**Soundness:** 2
**Presentation:** 2
**Contribution:** 2
**Rating:** 2
**Confidence:** 3

**Summary:**

This paper proposes PAFT (Parallel fine-tuning), a new paradigm for large language model (LLM) fine-tuning designed to alleviate the so-called alignment tax — the degradation in task performance caused by sequentially performing Supervised Fine-Tuning (SFT) followed by preference alignment (e.g., DPO, ORPO). Instead of training SFT and alignment in sequence, PAFT conducts them independently and concurrently on the same pretrained model, producing two LoRA-based adapters. After both trainings finish, the two sets of delta parameters are merged once through parameter fusion (using methods such as TIES or Task Arithmetic). The paper further introduces L1-regularization to sparsify the SFT adapter, arguing that sparsity reduces parameter interference during merging. Empirical results on the HuggingFace Open LLM Leaderboard and AlpacaEval show modest but consistent improvements over the sequential baseline.

**Strengths:**

- The paper recognizes a real issue: sequential SFT → alignment pipelines can cause performance degradation (alignment tax).
- Simple and practical idea — Running SFT and alignment in parallel is conceptually simple and can be implemented using existing adapter-based methods.
- Empirical improvement — Results on 7B and 70B models show consistent, reproducible gains (+4–5%) across multiple benchmarks and merging strategies.
- Systematic evaluation — Various merging methods (TIES, Task Arithmetic, etc.) are compared, and ablations on λ (L1 strength) are provided.

**Weaknesses:**

[Lack of direct evidence for “alignment tax reduction.”]
The central claim — that PAFT mitigates catastrophic forgetting — is supported only by indirect evidence (i.e., downstream accuracy). No explicit measurements of forgetting are provided. Hence, the causal link between “parallel training” and “reduced forgetting” remains unclear.

[Conceptual inconsistency in the notion of forgetting]
In PAFT, SFT and alignment are trained independently and merged only after both are finished. This design eliminates the sequential overwrite problem by removing the original SFT model rather than preserving it. Therefore, the framework does not truly prevent forgetting — it simply precludes the ability to measure or recover it. Author should clarify the notion of 'forgetting' that to be tackled in this article.

**Questions:**

[Clarify terminology.]
I think it might be better to replace “Parallel Training” with “Concurrent Fine-Tuning” or “Dual-Path Fine-Tuning” to reflect the conceptual design more accurately. The terminology of parrallel training have been widely used in multi-gpu training already.

[Provide direct forgetting analysis.]
Include explicit experiments measuring forgetting. This would substantiate the claim that PAFT mitigates catastrophic forgetting rather than merely achieving higher accuracy.

---

### Official Review · Reviewer_zros · 2025-11-03

**Soundness:** 3
**Presentation:** 3
**Contribution:** 3
**Rating:** 6
**Confidence:** 3

**Summary:**

This work proposes to replace the usual sequential SFT -> RLHF/DPO pipeline with a parallel training regime. The idea is to do both stages in parallel, and merge the resulting models. The paper focuses on the impact of model sparsity to model merging.

**Strengths:**

The authors argue that the interaction between the SFT and RLHF/DPO stages is underexplored, and cite catastrophic forgetting effects (alignment tax) - this is a very strong motivation for this work.

The evaluation covers a sufficient number of models and tasks and compares against fair baselines.

The approach is very straightforward which increases the chance of being adapted.

**Weaknesses:**

I am wondering about the convergence behavior of the approach: How much more brittle is it to do xPO directly on the pre-trained model? Is PAFT's training stability similar to sequential training?

On a similar note: How does your method compare in terms of computational complexity?

Since it is mentioned in the introduction, an analysis that shows that the alignment tax is indeed mitigated would be helpful.

To sum up: giving a bit more color to the whole approach beyond just reporting the final numbers (except Sec. 5.1) would make this a more round-up contribution.

Minor comments:
- The tables exceed the margins - can be fixed by nicer number formatting (eg 100*%.2f)
- Citations formatted without brackets

**Questions:**

- See weaknesses: Can you say more about the computational aspect and the convergence behavior?
- How sensitive is the lambda parameter? What model is shown in Table 4, and would it look similar for other models or do you need to re-tune lambda every time?

---

### Author Response · Authors · 2025-11-12
**No reviews released yet for this submission**

Dear Program Committee Members,

We want to kindly highlight that no reviews for this submission appear to have been released yet. Looking forward to your guidance.

Kind regards,
Authors.

---

### Meta-Review · Area_Chair_ZBMo · 2025-12-28

**Summary:**

This paper introduces PAFT (Parallel Fine-Tuning), a framework that replaces the conventional sequential SFT → preference alignment (e.g., DPO) pipeline with a concurrent training regime, followed by a single model-merging step. The work is motivated by the phenomenon of alignment tax, i.e., performance degradation caused by sequential fine-tuning. The paper further studies the role of adapter sparsity in improving merge quality. Across reviewers, there is agreement that the problem is well-motivated, the approach is conceptually simple and practical, and the empirical results show consistent improvements over sequential baselines on multiple models and benchmarks. At the same time, reviewers raised concerns regarding the conceptual framing of “alignment tax”, the lack of direct evidence for forgetting mitigation, increased training complexity, and clarity of presentation. As no rebuttal or author response was provided, these concerns remain unresolved and informed the final recommendation.

**Reviewer Concerns:**

1. Multiple reviewers noted that PAFT avoids sequential overwriting by construction rather than explicitly preserving knowledge. The paper does not provide direct measurements of forgetting or alignment tax, relying instead on downstream accuracy as indirect evidence. As a result, the causal claim that PAFT mitigates forgetting remains insufficiently substantiated.

2. Reviewers consistently requested explicit experiments quantifying performance drop before and after alignment. Without such analysis, the central motivation remains partially unvalidated.

3. PAFT requires training two separate adapters and performing a merge step. The paper does not provide a detailed comparison of total computational cost (FLOPs, wall-clock time, memory) relative to the standard sequential pipeline.

4. Several reviewers noted organizational issues and missing contextual discussion, especially around sparsity in alignment training and prior work. Some reported baseline discrepancies (e.g., unexpectedly low GSM8K numbers) were not clarified.

**Reviewer Scores:**

Reviewer zros: Likely unchanged
Reviewer qVn5: Likely unchanged
Reviewer LzpG: Likely unchanged
Reviewer mtHb: Likely unchanged

---

### Decision · Program_Chairs · 2026-01-26

Reject